# Lightweight Blockchain-Based Scheme to Secure Wireless M2M Area Networks

**Karam Eddine Bilami and Pascal LORENZ \***

IRIMAS/GRTC Institute, University of Haute Alsace, 68008 Colmar, France; karam-eddine.bilami@uha.fr
\* Correspondence: pascal.lorenz@uha.fr

**Abstract:** Security is a challenging issue for M2M/IoT applications due to the deployment, decentralization and heterogeneity of M2M and IoT devices. Typical security solutions may not be suitable for M2M/IoT systems regarding the difficulties encountered for their implementation on resource-constrained devices. In this paper, we discuss the architectures deployed for M2M communications and the security challenges, as well as the vulnerabilities and solutions to counter possible attacks. We present a lightweight design based on a private blockchain to secure wireless M2M communications at the device domain level. Blockchain integration provides secure storage of data while preserving integrity traceability and availability. Besides, the evaluation and experimentations under NS3 simulator of the proposed scheme show that the authentication mechanism is lightweight, and presents better performances comparatively to other protocols in terms of key parameters as communication and computational overheads, average delay and energy consumption.

**Keywords:** internet of things; M2M communication; authentication; M2M layer; blockchain

## 1. Introduction

Machine-to-machine (M2M) communication, also referred to as machine-type communications (MTC) were first used in telemetry applications for control and management of electrical, gas and water infrastructures through wired and direct communication between devices without human intervention.

M2M is considered as a concept somewhat similar to the IoT (Internet of Things). In fact, it is a type or subset of IoT communications like H2H (Human to Human) and H2M (Human to Machine). One difference is that M2M can use P2P or direct communication as well as communications over the Internet, but H2M and H2H connect devices with other devices or people only over the Internet.

M2M wireless communications present particular features since they support a large volume of M2M devices and transmit small amounts of data at regular or irregular intervals. Some M2M applications require low latency, high reliability and support various mobility profiles. Besides, they consume low power compared to other standards.

A definition that has large consensus [1,2] is as follows: M2M communications refers to any technology that enables the exchange of data between various devices (smartphones, health WPAN devices, sensors, embedded controllers, actuators, etc.) over wired or wireless networks, in order to perform a variety of actions without (or with limited) human intervention.

With the advent of wireless technologies (802.11 (Wi-Fi), 802.15.4 (Zigbee), radio frequency identification (RFID), ultra-wideband (UWB) radio technology, 802.15.1 (Bluetooth), 3G and 4G cellular networks, etc.), M2M applications have seen a growing interest. They are still expected to revolutionize the performance of various fields, especially with the arrival of 5G networks, which greatly improve connectivity between devices. In addition, the cost of M2M communication will decrease, bringing more opportunity and

benefits to industrial control, healthcare monitoring, critical infrastructure monitoring and maintenance, security and intelligent management of smart cities [3].

However, M2M communications face a number of challenges at different levels of M2M architecture as defined by the European Telecommunications Standards Organization (ETSI) [1].

At the low level of this architecture (i.e., at device network domain), communications take place between constrained M2M devices (sensors, smart objects …) and gateways or base stations as well as edge nodes depending on the adopted architecture. Due to such constraints (limited energy source, constrained computational capability, access control…), the management of a constrained device network as defined in RFC 7548, presents different types of challenges compared to the management of a traditional IP network.

A key challenge for M2M communications is related to the security of M2M devices, which are projected to operate without human intervention and are therefore defenseless. Moreover, the heterogeneity of the technologies and systems implemented on these devices lead to more vulnerability and made difficult their protection against various attacks (Spoofing, Deny of Service, etc.).

Most studies in this area are based on centralized and weighty security solutions, which are in best part inappropriate for constrained M2M devices.

Very few works tackle the authentication process with decentralized approaches at the device network level.

In this paper, a lightweight blockchain-based authentication and authorization mechanism is proposed to secure communications, especially at perception layer, namely M2M device domain. The remainder of this article is organized as follows: in the next section, a survey of different security protocols for wireless M2M communications is presented. In Section 3, after a brief description of a standard M2M communication architecture and its different components, the proposed scheme based on lightweight authentication protocol for M2M device domain is presented. The evaluation of performance parameters using experimental simulation is given in Section 4. Section 5 deals with security analysis of the proposed scheme. Section 6 discusses the obtained results and highlights the introduced advantages of the proposed scheme in comparison with other related protocols. Finally, Section 7 concludes this paper and suggests some future research directions.

## 2. Related Works

Security protocols for M2M are E2E (End to End) communications; initiated from M2M applications installed at application and device domain layers.

At high level, the architecture (application domain) uses various M2M/IoT communication protocols (CoAP, HTTP, SIP, MQTT, etc.). Generally, security solutions are based on classical security protocols in the internet like SSL/TLS, DTLS, IPSEC, and HIP (Host Identity Protocol).

At low level (device layer) of M2M and IoT architectures, different security protocols have been proposed based on classical symmetric cryptographic schemes, or lightweight asymmetric ECDH (Elliptic Curve Diffie–Hellman Key Exchange) [4], or based on lightweight operations addition, subtraction, hash function and XOR [5,6].

A large number of lightweight cryptographic algorithms such as HIGHT, LEA, and PRESENT [7,8] have been proposed in literature for resource-constrained devices like RFID tag, sensors, etc.

### 2.1. Traditional Security Protocols for Wireless M2M and IoT Communications

Many protocols have been proposed to secure communications between sensors, actuators and other devices for M2M device networks. Identity and authentication management of the interconnected devices can be maintained by a central authority founded on the Public Key Infrastructure (PKI). Unfortunately, methods based on IP-Sec, and SSL/TLS are inadequate for resource-constrained devices with limited computational resources.

Other studies [9,10] have focused on 6LoWPAN (IPv6 Low Power Wireless Personal Area Network) developed by IETF Internet Engineering Task Force in RFC 6568.

SPINS [11] is considered as the first security protocol that addresses various wireless security requirements, primarily at the device level. To encrypt messages and to generate message authentication codes, SPINS uses a lightweight version of RC5. It ensures security services separately over two protocols for unicast and multicast communications; however SPINS presents the forwarding problem.

TinySec [12], and Minisec [13] are authentication protocols operating, respectively, at link layer and network layer. They are considered as efficient authentication protocols, particularly well adapted for constrained nodes. Tinysec is the first fully implemented security protocol for WSNs, based on the message authentication code (MAC) and Cipher Block Chaining (CBC). Its drawback is related to the use of a single key to generate MAC codes.

Lightweight authentication and key distribution (LAKD) protocol for M2M communications is proposed in [7]. It permits a device to be sure of the identity of any other device before communication takes place. LAKD generates and distributes keys to be used to provide security services (confidentiality and integrity of exchanged data). Using basic operations that not require high processing resources (xor, addition, subtraction, etc.), LAKD is considered as an adequate protocol for devices with very limited resources.

In [8], the authors present lightweight authentication mechanism, based on hash and XOR operations, for M2M communications between a resource constrained industrial device (e.g., a smart sensor) comprising a secure element (SE) and a router comprising a TPM (Trusted Platform Module).

In [9], the authors propose a mutual authentication and key establishment for M2M communication in 6LoWPAN Networks. The protocol applies ECDH (Elliptic-curve Diffie–Hellman) algorithm to implement the secret key distribution among nodes.

Based on the analysis of EAKES6Lo protocol, showing its defect due to data loss after node capture, and its vulnerability to sinkhole attacks and plaintext chosen attacks, a new M2M communication mutual authentication protocol based on 6LoWPAN is proposed in [10]. The protocol builds up a sensible secret key dissemination mechanism and plans an anti-capture attack detection strategy for unattended nodes to stand up to attacks such replay attacks, sinkhole attacks, plaintext-chosen attacks, and physical capture attacks.

SAKES (Secure Authentication and Key Establishment Scheme) is proposed in [14]. It is based on lightweight public-key cryptography using ECC scheme in order to establish the session key. This one is established between the 6LoWPAN and the server, based on Diffie Hellman exchange. To decrease the computational load, related to cryptographic operation, on the sensor nodes, SAKES executes the computations on the gateway nodes and sends computed keys to the sensor nodes through the 6LoWPAN context.

Paper [15] proposes a three-factor authentication framework based upon a publish–subscribe pattern using message queue telemetry transport (MQTT) as application communication protocol. The general architecture supports mutual entity authentication between a remote user (subscriber), broker (gateway), and IoT node (publisher), based on password, identity and low-cost digital signature, using elliptical curve cryptography.

In paper [16], authors propose a secure user mutual authentication and key agreement protocol for smart home access. The proposed scheme combines physical context awareness and transaction history in order to improve the authentication and the security of the user remote access while avoiding the clock synchronization problem.

Paper [17] presents a signature-based authenticated key scheme to deal with the security issues in IoT. The mutual authentication between user and end device is verified using the Burrows–Abadi–Needham logic (also known as the BAN logic or broadly accepted BAN logic).

A three-factor authentication protocol for industrial internet of things with the aim to attain user secrecy is displayed in [18], the proposed protocol adopts a fuzzy extractor

that uses biometrics information in order to create strong keys, to be utilized as inputs to standard cryptographic techniques.

*2.2. New Approaches to Secure M2M and IoT Communications*

Centralized architectures suffer from Single Point of Failure (SPF). As an alternative solution, new decentralized approaches have been recently proposed to secure M2M communications.

Blockchain-based protocols are gaining popularity, and appear very promising to address the security challenges of M2M and IoT networks. Blockchain allows all transactions to be recorded in a shared database or a DL (Distributed Ledger). Hence, any modification can be easily detected. Theoretically, blockchain has the ability to provide security services as integrity, authentication and access authorization. However, there are challenges to overcome when used in M2M and IoT contexts due to its limited scalability and the constraints of the devices.

The authors in [19] use a blockchain-based approach to secure the routing process while considering blockchain as shared memory in WSN. In order to optimize and to secure the routing phase, the blockchain preserves track of the generated transactions on the network stating which nodes are transmitting and through which paths.

A blockchain-based information securing process, in which data is collected from the sensors utilizing an Unmanned Aerial Vehicle (UAV) as a relay, is presented in [20]. The collected information is safely stored within the blockchain on the mobile edge computing (MEC) server. Within the proposed scheme, the information is encrypted before transferred to the MEC server with the help of UAV. Upon getting the information, the MEC server approves the information and sender's identity. Effective validation is followed by storage of the information within the blockchain.

In [21], the authors present a blockchain based scheme in combination with a proxy re-encryption mechanism to guarantee the privacy of the information. The corresponding transactions are consequently managed through the agreed smart contract, and saved within the blockchain.

A decentralized blockchain architecture is adopted in [22] with the main purpose of facilitating secure information exchange within an APL (Adaptive Protection Platform).The architecture increases the throughput while leading to better scalability, and low node storage.

A lightweight identity management approach is presented in [23]. It is based on consortium blockchain for the IoT. The protocol addresses privacy and security issues in traditional centralized systems and centers on the concept of dispersing the authority of an identity system to a bunch of organizations. The proof of concept (PoC) as moderately lightweight consensus model is implemented.

A lightweight authentication scheme based on consortium blockchain is proposed in [24]. A cryptocurrency concept is used to build trust between entities. The proposed framework incorporates a PKI system to identify permissioned entities within the blockchain.

In [25], the proposed model provides a secure observed area for checking IoT data generated from IoT devices. Using the MAM (Masked Authentication Messaging) protocol, the encrypted environmental sensor data is broadcasted, stored, and fetched from a distributed ledger for the IoT, named IOTA to ensure integrity, security, and privacy.

**3. Proposed Scheme to Secure Device Layer Network**

M2M systems consist of a combination of connected systems and devices. To deal with M2M system architecture, we consider the overall architecture (Figure 1) given in ETSI Technical Report [1].

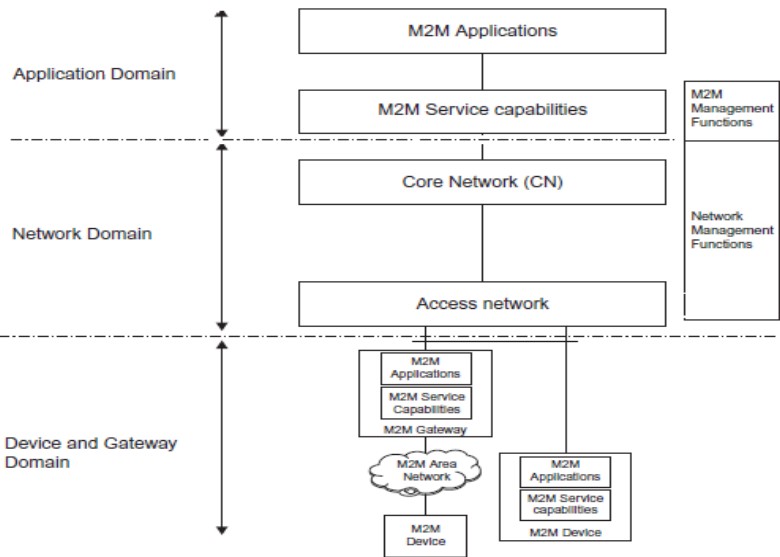

**Figure 1.** ETSI M2M system architecture—Functional view.

The general M2M architecture is divided into three interconnected domains called the M2M domain, the network domain and the application domain. The main constituents of this architecture can be described as follows:

- M2M devices for collecting and transmitting requested data.
- The M2M device network connecting the M2M devices to M2M gateways.
- M2M gateways interconnecting M2M device networks to communication networks.
- M2M communication network connecting M2M gateways to Internet or to the M2M application servers.
- M2M server providing data and ensuring different services from various M2M applications.

*3.1. Architecture of M2M Layer*

In the proposed architecture for the M2M device layer (Figure 2), device networks are linked to network layer via gateways depending on particular technologies (Wi-Fi, Zigbee, 4G and 5G cellular networks, etc.).

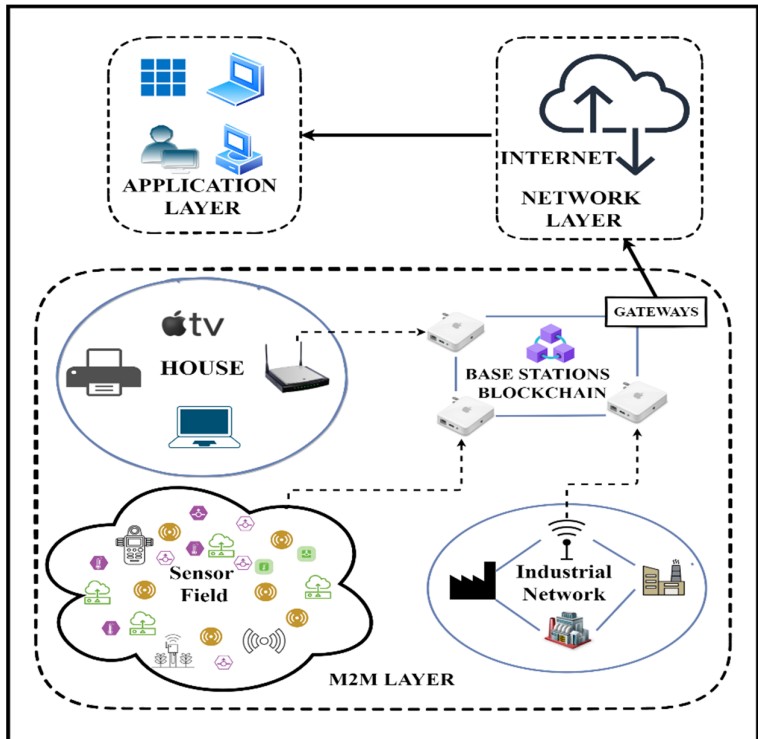

**Figure 2.** Architecture of M2M Layer.

The base stations (Figure 3) incorporating gateways are supposed with high (or unlimited) computational and energy resources.

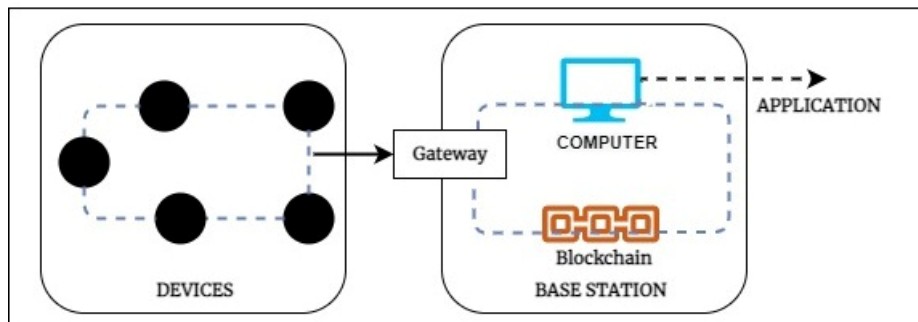

**Figure 3.** M2M base station.

They perform the following tasks:

- Registering a new node on the network as a new entity.
- Identity and authentication management of secure communications.

M2M nodes (sensors, smart objects, etc.) collect and transmit data packets before forwarding them to the M2M gateways. After receiving the packets, an M2M gateway manages the packets and provides appropriate paths for forwarding these packets to an M2M application server or to other device nodes on different device networks via wired/wireless networks. Of course, gateways perform appropriate protocol conversion and data segmentation or assembly (for example, from IP to 802.15.4 and vice versa).

### 3.2. Proposed Secure Communication Protocol

Blockchain technology as a distributed system allows avoiding Single Point of Failure (SPF) of centralized schemes. Blockchain permits us to tackle security concerns such as identity management, availability, and integrity and offers a solution for traceability to follow the history of each device activity while recording all its transactions.

Since constrained devices are unable to support the heavy processing that blockchain technology introduces for mining and cryptographic operations, a private blockchain is adopted and sited on a set of base stations closer to the sensed field (i.e., M2M area), which leads to more advantages, since it permits short delays and early control of devices. The blockchain is designed to contain all information related to each device. Figure 4 depicts the content and the block structure.

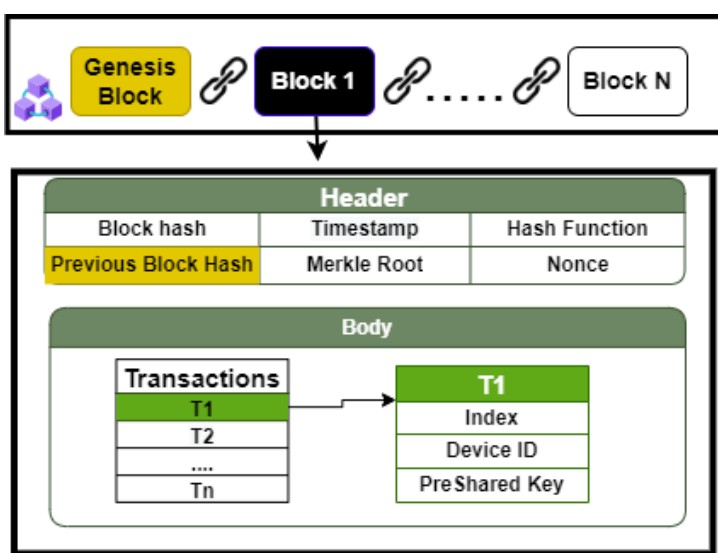

**Figure 4.** Blockchain block structure.

On the other hand, we choose Practical Byzantine Fault Tolerance (PBFT) to be implemented as lightweight consensus algorithm suitable for private blockchain. PBFT consensus guarantees data integrity and reliability even if N/3 -1 of the N network nodes are compromised. Thus, a new block is validated and added to the blockchain if (2 * N/3) +1 nodes, as a minimum, attain the consensus.

Before the communication takes place, the identification process is lunched with three phases: pre-registration, registration and authentication.

### 3.2.1. Pre-Registration Phase

A pre-shared key (PsK) used for registration of a device in Figure 5 can be generated by the M2M application server after verification of the nonexistence of the device in the blockchain. PsK is sent to the device via a secure channel, or automatically stored in the devices during the initial configuration of the device (prior to deployment) and communicated to the M2M application server via a secure channel.

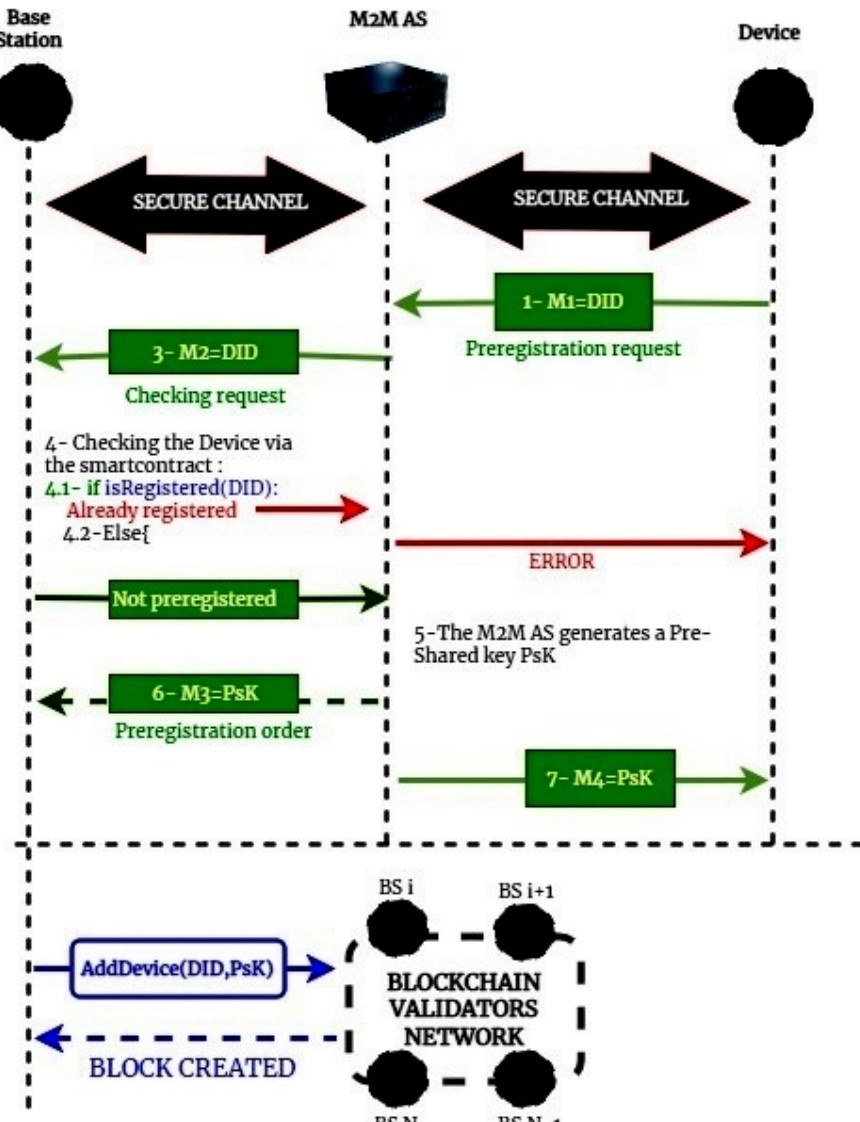

**Figure 5.** Pre-registration phase.

The device ID is transferred via a secure channel to be appended in the private blockchain using the deployed smart contract in Figure 6. First it checks the existence of the device in the blockchain using the function IsRegistered() from the smart contract, after that the node can be appended to the blockchain only if it does not already exist by calling the function AddDevice().

The registration order must be validated by the network nodes through PBFT consensus, so that it can be stored in the private blockchain. It is assumed that base stations are already registered at the network level.

---

**Contract: Authentication Scheme**

**Initialisation**:
- Event AddDev (uint256 index, uint256 DeviceID, uint256 PSK) // The arguments passed are stored in transaction logs once the event emitted, these logs are stored on blockchain.
- struct Device { uint256 DeviceID; string DeviceName;
- string DeviceType; uint256 PublicKey; }
- mapping( uint256 => Device ) public allDevices;
- uint256 public index;
**Functions**
Function 1: isRegistred(uint256){}
Function 2: AddDevice(uint256,uint256){}
Function 3: GetAllTheInfo(uint256){}
Constructor () public { index = 0; }

---

**Function 1: isRegistered**

**Input :** uint256 DeviceID
**Result:** returns a boolean // checks the existence of a device in the blockchain
If ( AllDevices[DeviceID].DeviceID!= 0 ) { return true; }
      else { return false; }

---

**Function 2: AddDevice // allows the addition of a device in the blockchain**

**Input :** uint256 DeviceID, uint256 PSK
**Result:** a new block is created (the device is added to the blockchain )
emit AddDev(index , DeviceID , PSK);
device = Device(DeviceID, PSK);
AllDevices[DeviceID] = device;
index = index + 1;

---

**Figure 6.** Authentication scheme smart contract.

Before starting the information exchange, the BS and end device must establish a session key (SK) to be used for symmetric encryption. An AES-128 bit is the most efficient block cipher (approved by the NIST) for confidentiality purpose in terms of security. However, its implementation on resource constrained devices is inappropriate due to its high execution cycles and memory consumption. Additionally, a lightweight derived variant of AES such as PRESENT [6] with shared key of 80 bits and block of 64 bits, presenting accepted performances, is suggested for ensuring confidentiality in our design.

The session key is obtained from the base point of the ECC curve used by the ECDH key exchange protocol to be transferred to the device for secure storage. The session key has a validity period and can be regenerated for each new session. Table 1 presents the used parameters with their lengths in bits.

**Table 1.** Parameter notations and lengths.

| Symbol | Description | Length (bits) |
|--------|-------------|---------------|
| DID | Device identity | 128 |
| DPbK | Device Public Key | 160 |
| DPrK | Device Private Key | 160 |
| BSPbK | Base Station public Key | 160 |
| BSPrK | Base Station private Key | 160 |
| PsK | Pre-shared key | 128 |
| SK | Session key | 80 |
| MAC | Message Authentication Code | 256 |
| Tstmp | TimeStamp | 32 |

The private keys DPrK and BSPrK are integers randomly chosen in the interval [1, n − 1], where n is the order of the curve. The public keys of the BS and device are correlated with their respective private keys.

Since: DPbK = DPrK* $\{\backslash\text{displaystyle } d\_\{A\}Q\_\{B\} = d\_\{A\}d\_\{B\}G = d\_\{B\}d\_\{A\}G = d\_\{B\}Q\_\{A\}\}$G and BSPbK = BSPrK*G where G is the base point on the curve, the shared secret key is then given by Equation (1):

$$\text{BSPrK*DPbK} = \text{BSPrK(DPrK* } \{\backslash\text{displaystyle } d\_\{A\}Q\_\{B\} = d\_\{A\}d\_\{B\}G = d\_\{B\}d\_\{A\}G = d\_\{B\}Q\_\{A\}\}G) = \text{DPrK(BSPrK*G)} = \text{DprK*BSPrK} \tag{1}$$

When a wireless device tries to connect to the network for the first time, the actual registration phase will be launched. The device starts by scanning the environment and requests the connection to the nearest base station.

3.2.2. Registration Phase

The registration phase is composed of five steps (Figure 7) detailed as follows:

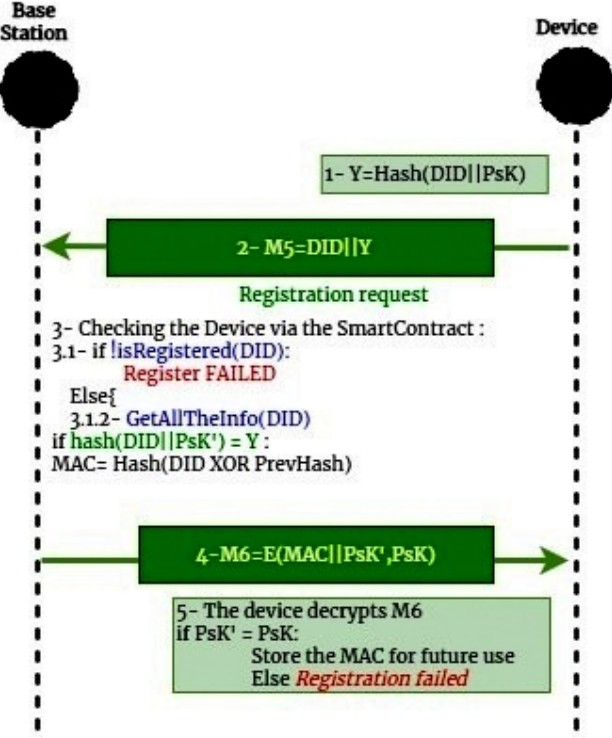

**Figure 7.** Registration phase.

Step 1: The device computes Y, the hash of (DID||PsK). A lightweight variant of SHA-3 with 256 bits is adopted as it is stated by the NIST, the most likely to be used in area of constrained designs.

Step 2: The device sends DID and Y to the base station.

Step 3: The base station checks the existence of the device in the blockchain, and if it does not exist, the registration is refused. In the other case, function 3, "GetAllTheInfo", in Figure 8 from the smart contract is called to get the device information. The BS computes hash(DID||PsK) and compares it to Y, and if the comparison holds, it computes the MAC as follows: MAC = Hash(DID XOR PH), where PH is the previous hash of the block, which contains the identifier of the device (if the device is stored in the block I, PH would be the hash of the block i−1).

Step 4: The base station encrypts the MAC and the PsK' using the PsK where PsK' is the device pre-shared key stored in the blockchain.

Step 5: The device decrypts M6 and checks if PsK' equals to its PsK, the MAC is then stored for future use if the comparison holds, or else the registration fails.

---

**Function 3:** GetAllTheInfo // returns the device information

**Input :** uint256 DeviceID
**Result:** Returns uint256, uint256 // the pre shared key and the previous block hash
Return (AllDevices[DeviceID].PSK , block.blockhash( AllowedDevices[DeviceID].index-1));

---

**Figure 8.** GetAllTheInfo function.

### 3.3.3. Authentication Phase

The authentication of any device takes place with a valid Message Authentication Code (MAC) because the smart contract is deployed on all base stations, and allows finding the hash of the precedent block that contains the identifier of the device. This mechanism brings more security to the network. The authentication phase is composed of seven steps (Figure 9) depicted as follows:

Step 1: The device generates its private key, and calculates its public key.

Step 2: The device computes Z where Z = Hash(Mac||PSK||DPbK).

Step 3: The device sends DID, Z, DPbK to the BS.

Step 4: The BS checks the timestamp (the value of the timestamp must be within an acceptable range of the current time). The BS after that checks if the device is already registered, if not, it cancels the authentication, else, it checks if the device was already authenticated (by checking its existence in the authenticated devices table), if it does not exist in the table, the BS collects the device information by calling the function GetAllTheInfo(DID) from the smart contract, then it computes X where X = DID XOR PH, after that, it checks whether Y is equal to hash(X||PsK'||DPbK||Tstmp), the authentication fails if the comparison does not hold, otherwise the device is successfully authenticated.

Step 5: The BS generates its private key, and computes its public key where BSPbK = BSPrK*G, and calculates the session key where SK = BSPrK*DPbK.

Step 6: The BS encrypts SK and BSPbK using the device public key, the result will be sent to the device

Step 7: The device decrypts M4 and computes the session key, the result will be compared to the session key received in M4, the BS is successfully authenticated if the comparison holds, otherwise the authentication fails.

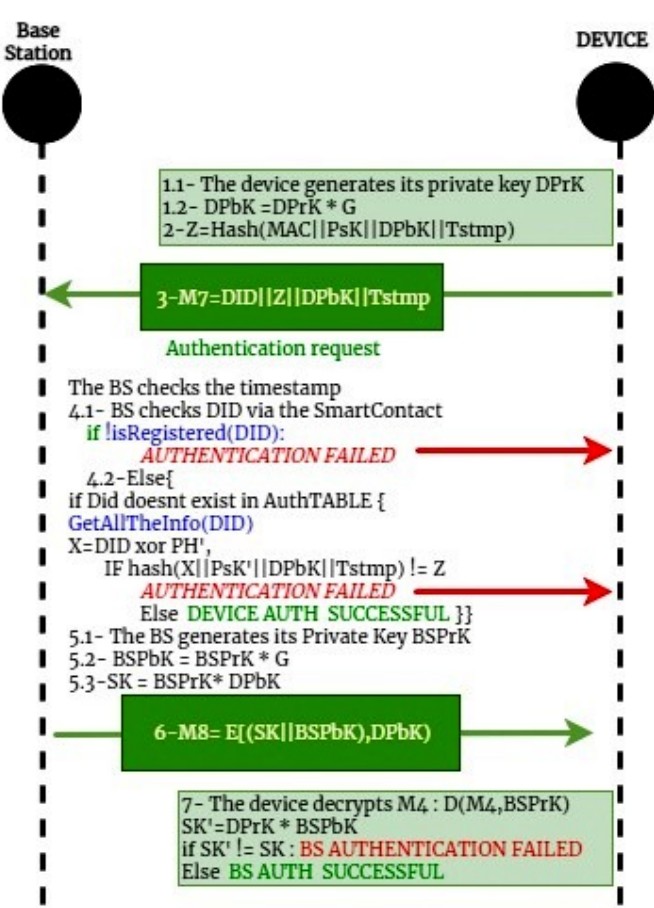

**Figure 9.** Authentication phase.

## 4. Performance Evaluation

In this section, we evaluate the performances of the proposed authentication mechanism in terms of key parameters as communication and computational overheads, average delay and energy consumption introduced by the authentication mechanism.

### 4.1. Communication Overhead

The exchanged authentication messages size (communication overhead) is an important parameter in the evaluation of performances; the communication overhead is defined as the sum of message lengths (in bits) transmitted and received by a device in the authentication process. Here, the protocol uses six messages to achieve the authentication. Communication overhead is calculated as follows:

$$M1 \ (128 \ bits) + M4 \ (128 \ bits) + M5 \ (384 \ bits) + M6 \ (384 \ bits) + M7 \ (576 \ bits) + M8 \ (240 \ bits) \tag{2}$$

Communication overheads of different protocols are given in Table 2. The proposed protocol presents a limited overhead storage and limited communication cost comparatively to other protocols as illustrated by the graph in Figure 10.

**Table 2.** Communication cost.

| Protocol | Communication Cost (bits) |
|---|---|
| Proposed protocol | 1840 |
| [8] | 1920 |
| [15] | 2560 |
| [16] | 2304 |
| [17] | 2528 |
| [18] | 2688 |
| [24] | 3152 |

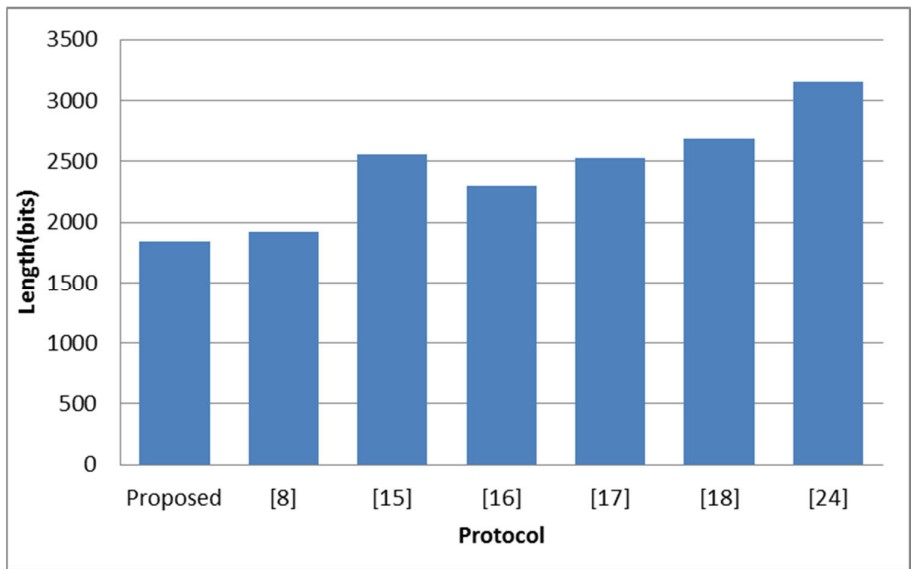

**Figure 10.** Communication overhead comparison.

### 4.2. Computation Overhead

The cost of computation overhead takes into consideration the needed time to perform different cryptographic operations (hash function, encryption and decryption and ECC arithmetic operations). The estimation of the computational cost is based on the same hypothesis utilized in [17] as shown in Table 3.

**Table 3.** Approximate time taken by each cryptographic operation [17].

| Symbol | Function | Computational Time (s) |
|---|---|---|
| T (h) | Hash function time | 0.00032 s |
| T (sym) | Symmetric encryption/decryption time | 0.0171 s |
| T (eca) | ECC point addition time | 0.0044 s |
| T (ecm) | ECC point multiplication time | 0.0171 s |

The obtained results in Table 4, show that the proposed scheme presents the lowest total computational cost (TCC) in Figure 11 including device computational cost (DCC), despite the fact that device computational cost is slightly higher comparatively to [18].

**Table 4.** Computational cost (CC) comparison in seconds.

| Protocol | Device CC (DCC) | DCC in Seconds | Total CC (TCC) | TCC in Seconds |
|----------|-----------------|----------------|----------------|----------------|
| Proposed | 2 T (h) + 2 T (sym) + T (eca) + T (ecm) | 0.0563 s | 5 T (h) + 4 T (sym) + T (eca) + 2 T (ecm) | 0.113 s |
| [15] | 3 T(ecm) + 1 T (h) + 2 T (eca) | 0.0604 s | 10 T (ecm) + 7 T (h) + 4 T (eca) | 0.190 s |
| [17] | 4 T (ecm) + 3 T (h) | 0.0693 s | 14 T (ecm) + 12 T (h) | 0.243 s |
| [18] | 4 T (h) + 2 T (sym) | 0.0354 s | 3 T (ecm) + 19 T (h) + 8 T (sym) | 0.194 s |

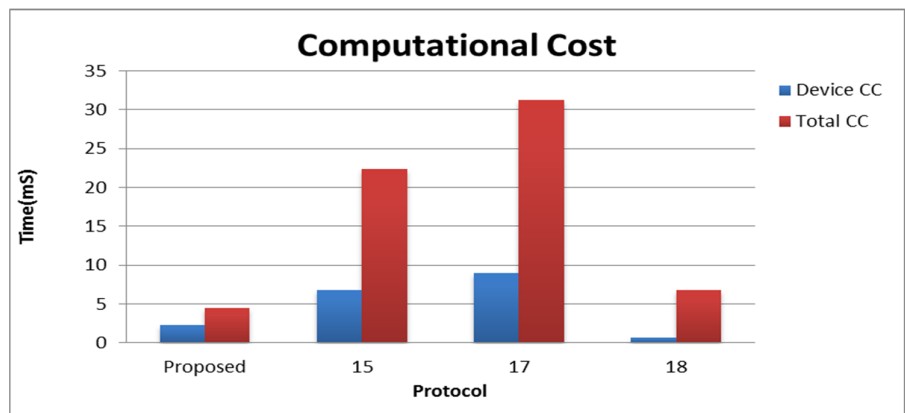

**Figure 11.** Computational cost comparison.

### 4.3. Delay Overhead

The Average overhead Delay (or mean delay) is an important metric to consider in order to evaluate the influence of the communication overhead, that is introduced by the proposed protocol for communication security. In our evaluation, this delay includes the MAC layer queuing besides transmission and propagation delays. It is expressed for one packet as the difference between the time of packet reception at the destination (rec_time) and the time of packet emission at the source side (emis_time).

The Average Delay Overhead (AoD) for a network of N devices is calculated as follows:

$$\text{AoD} = \sum_{j=1}^{N} ((\sum_{i=1}^{M} \text{rec\_time}(i) - \text{emis\_time}(i)) / M) / N \tag{3}$$

M: number of packets (overhead) introduced in the authentication process of device j.

### 4.4. Energy Consumption

The model described in [26] and known as first order radio communication model is adopted in our evaluation. Eelec is supposed to be the radio energy consumption for transmitting or receiving 1 bit of data, energy dissipation for transmitting 1 bit of data from source to destination node at the distance d can be estimated by the following equation:

$$E_{tx} = l \cdot E_{elec} + l \cdot E_{amp} \cdot d^2_{\text{toBS}} \tag{4}$$

The total energy dissipation by the transmission by a device of M messages:

$$E_{tx\_tot} = \sum_{i=1}^{M} l_i \cdot E_{elec} + l_i \cdot E_{amp} \cdot d^2_{j \text{ toBS}} \tag{5}$$

While the energy dissipation by a device to receive l bit data is formulated as:

$$E_{rx} = l \cdot E_{elec} \tag{6}$$

The total energy dissipation by a device for the reception of *M'* messages is:

$$E_{rx\_tot} = \sum_{i=1}^{M'} l_i \cdot E_{elec} \tag{7}$$

The total energy overhead consumed by the authentication process of N devices in our case, can be estimated by the following Equation (6):

$$E_{tot} = \sum_{j=1}^{N} (\sum_{i=1}^{M} l_i \cdot E_{elec} + l_i \cdot E_{amp} \cdot d_{j}^{2}{}_{toBS} + \sum_{i=1}^{M'} l_i \cdot E_{elec}) \tag{8}$$

where:

$l_i$: The total size in bits of transmitted data by the device.

$E_{elec}$: the dissipation of radio energy.

$E_{amp}$ is transmission amplifier energy dissipation.

$N$: The number of devices in the network.

The model is adequate for low distance. For longer distances ($d > d0$), the term $d^2$ is replaced by $d^4$ in Equations (4), (5) and (8). $d0$ is the crossover distance as defined in the radio communication model [26].

### 4.5. Simulation Setup and Parameters

To evaluate the performances of the proposed protocol in terms of average delay, packet delivery ratio and energy consumption, we have carried out simulations under NS3 simulator on Linux workstation. Base Station is located at the origin of the coordinate system. The smart devices are placed from 1 to 100 m from the BS (we place a device every two meters).

Communication is considered across the IEEE 802.15.4 LWPAN and 2.405 GHz (channel 11). The energy consumption for transmitter and receiver electronics is 50 nJ/bit and for transmitter amplifier is 100 pJ/bit/m². The other parameters are assumed to be at their default values as defined in NS3.

The simulation parameters are presented in Table 5.

**Table 5.** Simulation parameters.

| Parameter | Description |
|---|---|
| Platform | NS3.33/Ubuntu 20.04 |
| Programming | C++ |
| Transmit Power (dBm) | 0 dBm |
| Eelec | 50 nJ/bit |
| Eamp | 100 pJ/bit/m² |
| Maximum distance | 100 m |
| Number of devices | 50 |

From the experiments, considering an IEEE 802.15.4 M2M device network as case study, the average delay overhead, Packet Delivery Ratio (PDR) and energy overhead consumption of the network, obtained from the simulation are shown in Figures 12 and 13.

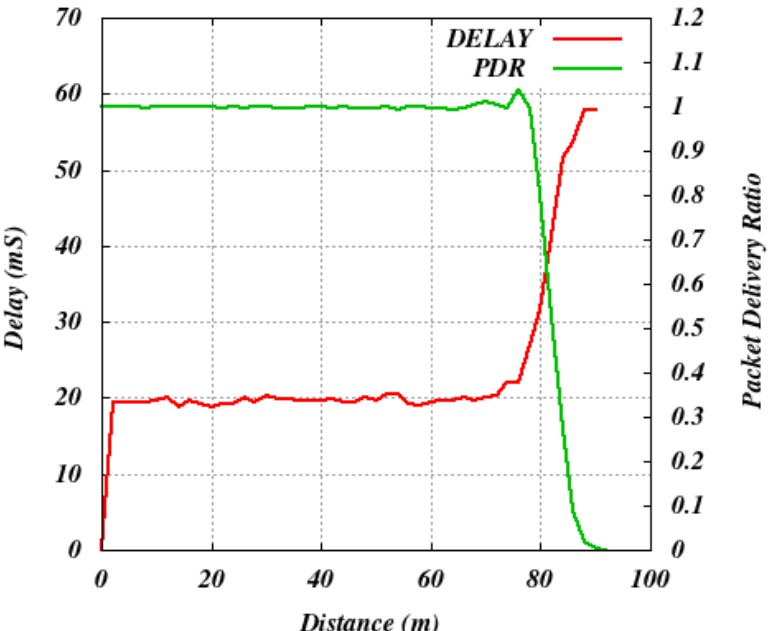

**Figure 12.** Average delay overhead, packet delivery ratio vs. distance.

The simulation results illustrated by the graph in Figure 12 show that the overhead delay increases with the evolution of the distance. In our experimentation, 92 m is the maximum distance covered by the device radio; over longer distances (d > 92 m), the packet delivery ratio (PDR) becomes equal to zero. The reason is that the transmission power of 0 db corresponds to a communication range of approximately 87 to 92 m.

The proposed protocol delivers comparable communication and storage overheads and PDR as the other studied protocols, but with relatively less energy consumption by wireless devices as shown in Figure 13. This leads consequently to an improvement of the lifetime of the M2M device network.

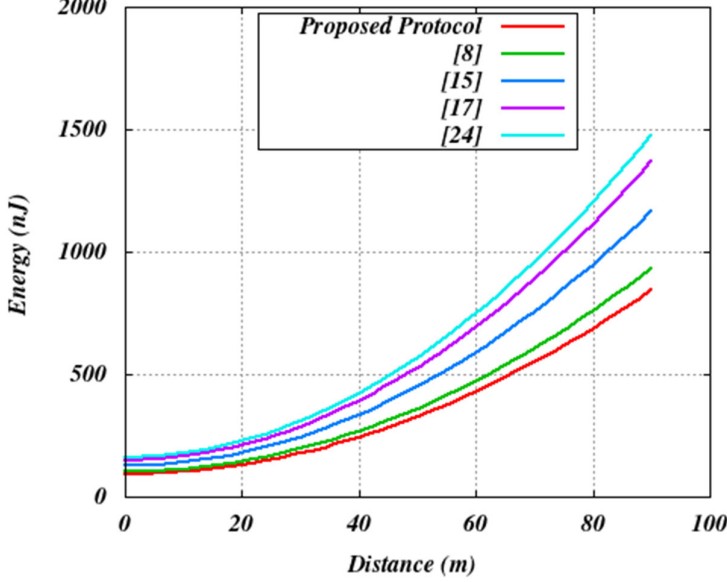

**Figure 13.** Energy overhead consumption.

## 5. Security Analysis

To check the validity of our scheme, informal analysis is conducted. We use a common adversary model [16,24] describing well-known security attacks at M2M device network level, and explain how the proposed scheme counters them. It is believed that a malicious attacker can secretly intercept communications or alter them between two entities. He can also impersonate another device identity in order to create fake requests and transactions, hinder the communication, discard transactions, delete or modify a transaction data, or link a user transaction to its identity. An adversary also may replay a stream of previously transmitted messages to the others. The adversary can also attempt to spoof the signature of a legitimate entity and sends it to others. Base stations are assumed to be secure.

(1) Man-in-the-middle attack: In addition to the mutual authentication provided by our scheme, the use of hash functions by the device during all the phases is considered as a proof of the device identity validity, each time the base station receives a message from the device, it retrieves its information from the blockchain and hashes them in order to compare the result to the received hash code, the comparison result shows if the data has been modified or not by a man in the middle.

(2) Replay attack: the adversary fraudulently delays or resends a request to the receiver. We expect that an authentic device has sent M5 to the base station, in case an adversary tries to impersonate the authentic device by replaying M5, the base station will dismiss the registration request since the device is already registered. In case an adversary tries to imitate the authentic device by replaying M7, the base station calculates the distinction between the received timestamp and its current timestamp, the base station disregards the received message in case the value is out of range.

(3) Impersonation attack: We consider that a device B intends to impersonate device A. However, Device B cannot obtain the MAC of device A. If device B wants to generate the MAC of device A itself, it will need the previous hash of the block where device A was stored in the pre-registration phase, this could not be possible because the devices have no access to the blockchain, and there is no function in the deployed smart contract, which returns a MAC except in the registration phase.

(4) Mutual authentication: Mutual authentication refers to two entities authenticating each other's identities before formal communication is established. Upon receipt of M7, the base station retrieves the device information from the blockchain and hashes them in order to compare the hash to Z received in M7. The device is considered authenticated if the comparison holds. When the device receives message M8, which contains BS public key and session key, previously calculated by the BS, the device computes the session key and compares it to the session key received in M8, the base station is also considered as authenticated if the comparison holds.

(5) Forgery Attack: the adversary attempts to spoof the signature of a legitimate entity and sends it to others. In the proposed scheme, the pre-shared key is exchanged in a secure canal at pre-registration phase. In the other phases, the PSK is always encrypted before any exchange. As long as it is not compromised, the adopted cryptographic theory ensures that it is infeasible to fake a valid signature without knowing the pre-shared key.

## 6. Discussion

Table 6 below, illustrates the strengths and weaknesses of different security protocols while highlighting the properties of the proposed blockchain-based scheme.

In addition to better performances at device level (i.e., low computational cost, low communication and storage overheads using relatively limited number of exchanged short messages), the proposal uses longer cryptographic keys of 160 bits, comparatively to other protocols using 128 bits hash values for cryptographic functions [8]. This leads to a more robust cryptographic solution (160 bit ECC public keys is equivalent to 1024-bit

RSA cryptosystem [17]). Compared to our scheme, the weak point of the scheme presented in [24] is its higher energy consumption, as shown in Figure 13.

The proposed scheme benefits from advantages offered by the blockchain technology, which leads to more powerful authentication process and more secure data sharing and storage while preserving their integrity, availability and traceability (or transaction history).

Most of proposed authentication protocols for securing constrained wireless devices ensure key agreement and mutual authentication, but present the drawback of SPF and centralized architecture (organized around an AS: Authentication Server as in [8], RA: Registration Authority in [16], GN: Gateway Node in [17], or Broker in [18]).

Contrarily, rather than using the authority of a centralized organization, the proposed scheme, taking advantage of blockchain technologies and comparable to the proposed system in [24], is decentralized and free of SPF. The security features of the blockchain evidently ensure a high level of trust without the need for managing trust amongst devices and base stations.

Obviously, the blockchain-based scheme consumes more processing time, related to the additional crypto mining and hashing operations, which consequently increases the power consumption at the base stations. The latter are considered high-resource devices, responsible for mining and storing the distributed ledger as a private and secure blockchain with relatively low additional computational and communication costs at the device network level.

**Table 6.** Comparison with other schemes.

| Protocol | Proposed | [8] | [15] | [16] | [17] | [18] | [24] |
|---|---|---|---|---|---|---|---|
| Centralized | No | Yes | Yes | Yes | Yes | Yes | No |
| Trust Level | High | Low | Low | Medium | Low | Low | High |
| Single point failure | No | Yes | Yes | Yes | Yes | Yes | No |
| Mutual Authentication | Yes | Yes | Yes | Yes | Yes | Yes | Yes |
| Key Agreement | Yes | Yes | Yes | Yes | Yes | Yes | Yes |
| Resist Man in the middle | Yes | No | Yes | Yes | Yes | No | Yes |
| Transaction History | Yes | No | No | Yes | No | No | Yes |

### 7. Conclusions

In this paper, we proposed a secure architecture for M2M layer communications. A security mechanism is suggested for mutual authentication between base stations and end devices once registered. A blockchain technology is used to provide security for M2M communications while improving the identity management and facilitating traceability, integrity and availability of data transactions.

The proposal is based on lightweight procedures involving limited overhead storage and a limited number of exchanged messages for identity management. Moreover, the security analysis of our proposal confirms that our proposal resists different attacks. As future work, we project to extend the proposed scheme in order to support mobility of devices and heterogeneity of networks. Moreover, we plan to evaluate through simulations the performance of the extended blockchain-based scheme, in which devices need to access the authentication service across high levels of the M2M architecture, and implement a new adapted PBFT lightweight consensus.

**Author Contributions:** Writing—original draft, K.E.B. and P.L. All authors have read and agreed to the published version of the manuscript.

**Funding:** This research received no external funding.

**Institutional Review Board Statement:** Not applicable.

**Informed Consent Statement:** Not applicable.

**Data Availability Statement:** Not applicable.

**Conflicts of Interest:** The authors declare no conflict of interest.

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
