# Peer review of "Lightweight Blockchain-Based Scheme to Secure Wireless M2M Area Networks"

_futureinternet, doi:10.3390/fi14050158_

Round 1

Reviewer 1 Report

The paper present a secure communication protocol based on a private blockchain to secure wireless M2M area networks. The description of the network architecture as well as the technical procedures of each protocol phase is clear. Analysis and simulation of the protocol is also presented. 

Some comments: 

1. In the evaluation section (and the simulation section), the proposed protocol is compared with other work ([8], [20-24]). However, only [8] is introduced in the related work section. The protocols in [20-24] should be explained to show the difference from this work. 

2. Several new decentralized approaches [15-19] to secure M2M and IoT communications are presented in Section 3.2. All of these are based on blockchain. The difference between the new protocol and these existing work should be included. Is there any reason these are not included in the evaluation section? In addition, if possible, these should be included in Table 6 (as those currently listed are all centralized except for the proposed one). 

3. In the proposed protocol, the private blockchain is hosted and maintained in a set of base stations closer to the sensed field. To make it secure, a certain number of such base stations are needed. Some discussion on the base stations as well as the overhead running the blockchain on these base stations should be included. In the current paper, evaluation and simulation is only limited to the communication between devices and the base stations (or M2M authentication servers). 

Author Response

Comments:

1.    In the evaluation section (and the simulation section), the proposed protocol is compared with other work ([8], [20-24]). However, only [8] is introduced in the related work section. The protocols in [20-24] should be explained to show the difference from this work.

Response 1:

Yes, according to this valuable comment, the different centralized schemes [20,21,23,24] compared, in the evaluation section to our proposal in terms of communication and computational overheads, have been introduced and explained in section 2 (related works) in the revised version.

2.    Several new decentralized approaches [15-19] to secure M2M and IoT communications are presented in Section 3.2. All of these are based on blockchain. The difference between the new protocol and these existing works should be included. Is there any reason these are not included in the evaluation section? In addition, if possible, these should be included in Table 6 (as those currently listed are all centralized except for the proposed one).

Response 2:
in respect of the reviewer comment, paper [22] dealing with centralized architecture has been replaced by a new paper [24]
which deals with a decentralized architecture for comparison purpose as recommended .

Really, most of blockchain based security protocols in the literature consider the performance of blockchain implementations in terms of cryptographic and mining complexity and consensus time consuming. Very few works tackle the authentication process with decentralized approaches at the device network level.
[15] proposes a blockchain-based approach to secure the routing process while considering blockchain as shared memory to support network status in real-time in order to optimize and secure the routing phase.
[16] deals with a blockchain based data acquisition scheme that facilitates the collection of data from loTs using UAV. After collection, the data are stored securely in blockchain at mobile edge computing (MEC) server
In [17] the authors propose a blockchain based scheme in combination with a proxy encryption mechanism to ensure the confidentiality of the data

Paper [18] presents a decentralized blockchain architecture that facilitates secure information exchange within an APL (Adaptive Protection Platform).
Paper [19] deals with an authentication protocol, but it focuses on the blockchain consensus side and neglects some performance parameters as the communication time overhead and energy optimization at device level.

3.    In the proposed protocol, the private blockchain is hosted and maintained in a set of base stations closer to the sensed field. To make it secure, a certain number of such base stations are needed. Some discussion on the base stations as well as the overhead running the blockchain on these base stations should be included. In the current paper, evaluation and simulation is only limited to the communication between devices and the base stations (or M2M authentication servers).

In this paper, we are interested in authentication procedure at device domain level. Since constrained devices are unable to support the heavy processing that blockchain technology introduces for mining and cryptographic operations, it is suggested that the blockchain should be hosted at base stations which are considered as high-resource machines supporting mining operations and storing the distributed ledger. In this work, we focus on the consumed overhead time and energy by the devices. Extra domain communications between base stations are assumed to be secure and are beyond the scope of this paper.

Reviewer 2 Report

In this paper, the authors present a private blockchain based secure design for M2M communications at the device domain level. The proposed blockchain is used for secure data storage that maintains integrity, traceability, and availability.

The proposed method is interesting, and holds merit. However, some points need to be addressed before the manuscript is ready for publication.

Major Comments:

The security analysis section, nor any other section, did not provide proper attack modelling. It analyzed the resistance to these attacks, but the attack models were not presented and the reader does not get adequate understanding of how these attacks take place, and thus how the proposed system counters them. These can be added as an independent section or subsection as well.

Table 6 is quite informative in terms of feature comparison of the proposed system with related works. However, the comparison lacks the performance measures details. Comparing the features is informative, but not enough. It is important that the reader sees a clear comparison of performance with previous works to be able to understand the superiority of the proposed system. Some of these parameters were mentioned in table 2. However, discussing these results was not included in the discussion section. 

In general, the discussion section did not really present meaningful insights into the importance of the results in comparison to previous works. I suggest expanding the section to detail the advantages of the proposed system compared to the related works.

Minor Comments:
1. The paper language needs review to improve readability and eliminate grammatical errors.
2. Proposed work section is expected to be section 2, while the network design is expected to come after the related works.
3. The conclusions can also include some suggested future research directions.

Author Response

Major Comments:

1.    The security analysis section, nor any other section, did not provide proper attack modelling. It analyzed the resistance to these attacks, but the attack models were not presented and the reader does not get adequate understanding of how these attacks take place, and thus how the proposed system counters them. These can be added as an independent section or subsection as well.

Thank you for this observation. Yes, we agree that the description of the different attacks was omitted in the first version. A common attack modeling (used in different referenced papers and dealing with the topic), which describes well-known attack approaches has been added in the security analysis section.

2.    Table 6 is quite informative in terms of feature comparison of the proposed system with related works. However, the comparison lacks the performance measures details. Comparing the features is informative, but not enough. It is important that the reader sees a clear comparison of performance with previous works to be able to understand the superiority of the proposed system. Some of these parameters were mentioned in table 2. However, discussing these results was not included in the discussion section.
In general, the discussion section did not really present meaningful insights into the importance of the results in comparison to previous works. I suggest expanding the section to detail the advantages of the proposed system compared to the related works.

We agree with this constructive comment. More discussions and comparison details have been added in the discussion section.

Minor Comments:
1.    The paper language needs review to improve readability and eliminate grammatical errors.
The paper has been revised. The authors have verified and made proofreading of the first version for more clarity. Grammatical errors have been removed. The quality of the paper has been improved.

2.    Proposed work section is expected to be section 2, while the network design is expected to come after the related works.

The paper has been reorganized, the standard M2M communication architecture and its different components in addition to the network design is relocated after the related works section.

3.    The conclusions can also include some suggested future research directions. Some future research directions have been added to the conclusion section as recommended.

Round 2

Reviewer 1 Report

The revised version addressed most of previous concerns. The authors are suggested to proofread the paper thoroughly before publication.